# Coriander (*Coriandrum sativum*) Cultivation Combined with Arbuscular Mycorrhizal Fungi Inoculation and Steel Slag Application Influences Trace Elements-Polluted Soil Bacterial Functioning

**DOI:** 10.3390/plants12030618

**Published:** 2023-01-31

**Authors:** Julien Langrand, Anissa Lounès-Hadj Sahraoui, Jérôme Duclercq, Robin Raveau, Frédéric Laruelle, Valérie Bert, Natacha Facon, Benoît Tisserant, Joël Fontaine

**Affiliations:** 1Unité de Chimie Environnementale et Interactions sur le Vivant (UCEIV-UR 4492), Université Littoral Côte d’Opale, SFR Condorcet FR CNRS 3417, CS 80699, 62228 Calais, France; 2Unité Écologie et Dynamique des Systèmes Anthropisés (EDYSAN UMR CNRS 7058 CNRS), Université de Picardie Jules Verne, UFR des Sciences, 80029 Amiens, France; 3Institut National de Recherche pour l’Agriculture, l’Alimentation et l’Environnement (INRAE), UMR Santé et Agroécologie du Vignoble (SAVE), Bordeaux Sciences Agro, ISVV, 33882 Villenave d’Ornon, France; 4Unité Technologies Propres et Economie Circulaire, INERIS, Parc Technologique Alata, BP2, 60550 Verneuil en Halatte, France

**Keywords:** steel slag, phytomanagement, trace elements, coriander, arbuscular mycorrhizal fungi

## Abstract

The cultivation of aromatic plants for the extraction of essential oils has been presented as an innovative and economically viable alternative for the remediation of areas polluted with trace elements (TE). Therefore, this study focuses on the contribution of the cultivation of coriander and the use of arbuscular mycorrhizal fungi (AMF) in combination with mineral amendments (steel slag) on the bacterial function of the rhizosphere, an aspect that is currently poorly understood and studied. The introduction of soil amendments, such as steel slag or mycorrhizal inoculum, had no significant effect on coriander growth. However, steel slag changed the structure of the bacterial community in the rhizosphere without affecting microbial function. In fact, *Actinobacteria* were significantly less abundant under slag-amended conditions, while the relative proportion of *Gemmatimonadota* increased. On the other hand, the planting of coriander affects the bacterial community structure and significantly increased the bacterial functional richness of the amended soil. Overall, these results show that planting coriander most affected the structure and functioning of bacterial communities in the TE-polluted soils and reversed the effects of mineral amendments on rhizosphere bacterial communities and their activities. This study highlights the potential of coriander, especially in combination with steel slag, for phytomanagement of TE-polluted soils, by improving soil quality and health.

## 1. Introduction

Soils provide fundamental ecosystem functions and play an important role in water and nutrient cycling, food production and the provision of renewable materials. Their ability to store carbon is essential for coping with climate change, and soil biodiversity is critical for soil fertility and biodiversity in general. Unfortunately, increasing pollution has made large areas of soil unsuitable for cultivation, a hazard to wildlife and humans, and uninhabitable for many living microorganisms. It is widely recognized that trace elements (TE), such as lead (Pb), cadmium (Cd), and zinc (Zn), can inhibit biological processes in ecosystems and affect the biodiversity, abundance, activity and function of soil microorganisms. These pollutants originate from human activities and, in particular, from smelters that have contributed to the production of significant amounts of dusts, resulting in severe contamination of soils with TE [1]. They represent an increased toxicity for microorganisms living in the soil and are also responsible for soil acidification, small size of soil aggregates and poor soil structure due to the lack of organic matter (OM), humic acids and, by extension, nutrients [2,3,4]. Therefore, it is crucial to reduce TE-toxicity in polluted soils to restore soil ecosystem services. Phytomanagement technologies, such as phytostabilisation, are suitable options to manage agricultural soils polluted with TE ecologically and economically [5,6,7]. Therefore, a crucial point for successful phytomanagement is the selection of TE-tolerant plant species that grow rapidly and produce enough biomass to allow recovery [6,8]. Previous studies have shown that the cultivation of aromatic plants, such as coriander (*Coriandrum sativum*), should be a promising option for phytomanagement of TE-polluted soils [6,9]. Indeed, this aromatic belonging to the *Apiaceae* family is cultivated for the production of essential oils (EO), which are an important way of valorizing plant biomass. The EO produced on TE-polluted soils have antibacterial, antifungal and antioxidant activities, and can be used for various applications [10]. Although it has been demonstrated that coriander can grow in TE-polluted soils, this plant species accumulates Zn and Cd in its aboveground parts, which makes it impossible to recycle residues from the distillation of coriander (shoots or seeds), in the methanization process, or as compost [6,9,11].

For example, in the context of circular economy, the incorporation into the soil of mineral basic amendments could improve soil parameters, such as pH, nutrient mobilization or soil structure and microbial composition [12,13,14]. In addition, mineral basic amendments, such as slags, can contribute to mineral nutrition and thus improve crop yields, but also play an important role in stabilizing soil aggregates and resistance to plant diseases [12,13,15]. Some studies have shown the role of slag in TE immobilization, especially Cd, Zn and Pb [13,16,17]. Furthermore, steel slag has been described to increase microbial biomass, soil enzyme activity and the number of copiotrophic bacterial taxa, and is believed to provide beneficial ecosystem services to plants [14].

Therefore, an adequate supply of mineral, organic or biological amendments could be interesting, first to reduce TE-mobility and bioavailability, and also to improve the growth of coriander and soil function [11].

Biological amendments, such as arbuscular mycorrhizal fungi (AMF) inoculum, have also been shown to be useful for plant protection. Indeed, AMF enable host plants to grow vigorously under stressful conditions, especially in TE-polluted soils [18]. They contribute significantly to the uptake of soil nutrients, especially phosphorus [19], but also nitrogen, calcium, sulfur, potassium and zinc, thereby increasing plant biomass and improving plant tolerance to soil pollution [20,21]. In addition, AMF produce glomalin, an insoluble glycoprotein that has soil binding properties, but also melanin and chitin, which reduce metal bioavailability by metal immobilization into hyphae [18,22,23]. Soil microbial communities are indeed crucial for soil structure, OM degradation and C cycling [3,24]. They are often used to assess the degree of soil remediation [8,25].

Many recent studies have described a synergistic effect of AMF inoculation and biological and mineral amendments (olive residues, biochar, phosphate sludge from marble waste, compost, manure) on plant growth in TE-polluted soils [26,27,28,29,30]. So far, there are few scientific works dealing with the combined application of AMF and mineral amendments (steel slag), and their effects on microbial communities and their activities are generally poorly studied [16].

Therefore, under *in situ* conditions, this study focuses on the potential benefits of adding amendments (mineral amendment alone or in combination with AMF inoculum) to improve coriander growth and microbial function of historically TE-polluted soils. To this end, we investigated plant growth parameters, soil bacterial biomass (phospholipid fatty acid, PLFA), the functional activity (community-level physiological profile) and structure (16S rRNA metabarcoding) of rhizosphere bacterial communities.

## 2. Results

### 2.1. Coriander Growth Parameters 

Although the plants had the same height after 11 weeks of cultivation regardless of the modality, we found that plants inoculated with AMF had a lighter aboveground dry weight than those non-inoculated (Table 1). In addition, the type and dose of the amendment did not alter the dry weight of coriander.

### 2.2. Mycorrhizal Rate

The mycorrhizal rate of coriander roots ranged from 31.7 to 51%, and was significantly higher in plots amended with 1.5 t ha^−1^ oven slag (51%) than in plots amended with 1.5 t ha^−1^ ladle slag (31.7%). No difference was observed between NI and I conditions, indicating the presence of native AMF. The dose of amendment did not affect the mycorrhization rate.

### 2.3. Soil Microbial Biomass

Total bacterial biomass was not significantly different between the two unamended conditions. Analysis of variance (ANOVA) of the Gram-positive bacteria biomass showed an increase with the lower dose of amendment (1.5 t ha^−1^) and with a combined effect of dose (1.5 t ha^−1^) and vegetation (Table 2). Moreover, the biomass of Gram-positive bacteria was significantly higher (8.14 µg g^−1^ of dry weight (DW) of soil) in soil amended with 1.5 t ha^−1^ oven slag and under vegetated conditions than under the same non-vegetated conditions (2.54 µg g^−1^ of DW of soil) (Table 2).

A significant effect of vegetation was observed on Gram-negative bacterial biomass. In fact, this increase was significantly higher in vegetated soils and NI condition supplemented with 1.5 t ha^−1^ ladle slag (5.30 µg g^−1^ soil) than under unvegetated modality (2.31 µg g^−1^ soil, Table 2).

The total bacterial biomass followed the same trend as the Gram-positive bacterial biomass. Indeed, a significant increase in total bacterial biomass was observed in the amended soil with 1.5 t ha^−1^ oven slag and under vegetated conditions (11.92 µg g^−1^ of DW soil) than in the same non-vegetated plots (5.05 µg g^−1^ of DW soil). Examination of the ratio of the biomass of Gram-positive to Gram-negative bacteria showed no significant difference between diverse modalities (Table 2).

### 2.4. Bacterial α-Diversity of Soil Biotopes 

To further investigate the effects of slags, vegetation and AMF inoculation on rhizospheric bacterial communities, the bacterial OTUs (Operational Taxonomic Units) were taxonomically assigned. The bacterial OTUs in the soil were assigned to 993 genera belonging to 33 phyla. *Actinobacteria* was the most represented phylum in each modality and accounted for between 41 and 67% of the total bacterial abundance (Figure 1).

Bacterial α-diversity was calculated independently for each modality. According to the Chao1 index, soil bacterial species richness remains stable across modalities (average 325.3). Similarly, both bacterial richness and diversity (Shannon index) were not affected by the addition of slag, mycorrhizal inoculation or vegetation (Table 3).

*Actinobacteria* associated with *Proteobacteria*, *Gemmatimonadota*, *Myxococcota* and *Acidobacteriota* accounted for over 85% of bacterial OTUs. In general, we found that the abundance of the different bacterial phyla, and thus the structure of the bacterial community, was affected by the conditions tested. One of the most important changes concerned the high abundance of *Actinobacteria*, which was even more pronounced under the vegetated conditions, than under the unvegetated conditions, except for the not inoculated and unamended condition (Appendix A). In contrast, a decrease of the *Gemmatimonadota* and *Acidobacteriota* abundances was observed under all vegetated conditions (Figure 1).

In addition to these changes in the abundance of the bacterial phyla, we also noted changes within phyla (Figure 2). For example, within the phylum *Actinobacteria*, we found that the genera *Pseudarthrobacter*, *Arthrobacter* and *Streptomyces* were enriched under vegetated conditions, whereas the genus *Gemmatimonadota* was favored under non-vegetated conditions.

### 2.5. Metabolic Potential of Soil Microbial Communities 

To understand the impact of taxonomic changes on the functionality of these microbial communities, a study was conducted on their ability to degrade different carbon sources. Overall, the metabolic potential of the soil microbial communities of each sample, expressed as average well color development (AWCD), was significantly higher under vegetated conditions than under the unvegetated conditions. The lowest value was found under the not inoculated, unvegetated and oven slag amended condition at 45 t ha^−1^. In addition, there was no significant effect of inoculation, type of amendment or dose on AWCD level (Table 4).

A multivariate ANOVA showed that functional richness was significantly higher in vegetated conditions than in the unvegetated conditions. Neither the inoculation, nor the type or dosage of amendment had a significant influence on the values of functional richness (Table 4).

## 3. Discussion

The cultivation of aromatic plants for the EO extraction has been presented as an innovative and economically viable alternative for the phytoremediation of TE-polluted areas [6]. However, the effects of growing aromatic plants, such as coriander, in combination with mineral (steel slag) and microbial (AMF) amendments introduced into the soil on the structure and potential function of bacterial communities in the TE-polluted rhizosphere, have never been studied. Thus, our results showed that coriander was able to establish and develop well on heavily TE-polluted soil, confirming previous work [6]. The addition of steel slag had no overall effect on plant growth. This result could be explained by the low nutrient requirement of coriander, but also by the OM content of this agricultural soil [6,31]. Consequently, it is likely that the slag did not provide any supplemental nutrient gain to the plant. Indeed, the beneficial effect of slag is controversial. Some studies have shown that these mineral amendments could contribute to mineral supply and consequently improve crop yields, while others showed no change or diverse responses, depending on the experimental conditions [13,15,32,33,34].

We have shown that coriander was able to form a mycorrhizal symbiosis with the native AMF present in our TE-polluted soil. Therefore, no significant difference in root mycorrhizal colonization rates was found between NI and I conditions, while AMF inoculation had no significant effect on shoot growth. In contrast to our results, some studies have shown that the introduction of AMF inoculum into TE-polluted and unpolluted soils can lead to a significant improvement in overall mycorrhizal rates, indicating different responses in experiments depending on the plant species, but also on the type of mycorrhizal inoculum [6,35,36]. We cannot exclude that the natural presence of native AMF in the TE-polluted soils masked the contribution of the exogenous mycorrhizal inoculum on coriander growth. In contrast, an increase in mycorrhizal colonization rates was observed in the presence of oven slag. This increase could be due to a reduction in TE-bioavailability or changes in pH or soil texture, as suggested by Hu et al. [16].

Although no benefits were found for the growth of coriander after mineral amendment and inoculation with AMF, several studies reported the crucial role played by these mineral and microbial amendments, as well as vegetation, in restoring soil functions that might be disturbed by TE-pollution [16,26]. In our results, the high bacterial diversity and richness indexes were comparable to some indexes obtained on unpolluted soils [37,38]. Moreover, despite the soil TE pollution, the functionality of the AWCD-based microbial communities indicated that the studied soils had a high metabolic capacity, which is typical of functional soils [39,40,41,42]. However, numerous studies have also shown a reduction in the abundance, activity, and diversity of soil microbial communities following TE exposure [42,43,44]. For example, Fontaine et al. [45] demonstrated a relatively low ratio of fungi-to-bacteria in this agricultural polluted soil, indicating an extensively managed soil in combination with tillage, high fertilization and a low C:N ratio favored bacteria. In this study, more Gram-positive than Gram-negative bacteria were observed under all conditions. Adaptation to an environment rich in TE leads microorganisms to exhibit activities of biosorption, bioprecipitation, extracellular sequestration, transport mechanisms and/or chelation [46]. Gram-positive bacteria are known for their diverse physiological and metabolic properties that allow them to thrive in a wide range of environments. Compared to Gram-negative bacteria, they appear to accumulate higher TE-concentrations on their cell walls [47]. Therefore, these bacteria are able to adapt easier to adverse conditions, such as pollution, and should even be used for TE-removal, which could explain their frequent presence in this soil [44,48]. Specifically, *Actinobacteria*, *Proteobacteria* and *Gemmatimonadota*, were the most abundant bacterial phyla in our TE-polluted soils. The *Actinobacteria* phylum has been highlighted for its TE-tolerance and even investigated for potential applications in remediation [49]. It plays a role in the degradation of polysaccharides, e.g., cellulose, xylan and chitin, OM turnover and in nutrient cycling [50]. *Proteobacteria*, the second most abundant phylum, is known to contain a wide range of functional bacteria, including fast-growing bacteria and many plant promoters, as well as growth-promoting bacteria [51]. Furthermore, *Proteobacteria* taxa have been described to be more abundant in heavily TE-polluted soils [44,52]. The latter phylum, *Gemmatimonadota*, is widespread in ecosystems around the world, making it an important member of soil bacterial communities [53]. Therefore, the long-term response of the bacterial community to TE results from the ability of these microorganisms to develop TE-tolerance without affecting overall community structure, as proposed by Brandt et al. [54].

The addition of amendments, such as steel slags or mycorrhizal inoculum, altered the structure of the rhizosphere bacterial community without affecting its metabolic potential, which was assessed by the AWCD parameter. In fact, the low dose of the mineral amendment increased bacterial biomass and caused a shift in the bacterial community structure. The *Actinobacteria* phylum was significantly less abundant under the slag-amended conditions, while the relative proportion of *Gemmatimonadota* increased. These results are in agreement with those of Gremion et al. [40] on a TE-polluted soil, but also with the functional redundancy of these communities [55]. Mineral granulometry has been shown to act as a physical support for the fixation of microorganisms (i.e., bacteria, fungi, etc.) and plants, as well as a nutrient reserve, and that the abundance of *Acidobacteria* decreases with increasing pH [56]. In this respect, and due to the alkaline nature of slags, the pH and mineralosphere can be important factors influencing microbial communities [56,57].

Cultivation of coriander also resulted in a change in the bacterial biomass and community structure. A significant increase in Gram-negative bacterial biomass under vegetated conditions and in the proportion of *Actinobacteria* phylum was detected. It is well known that plants secrete specific root exudates to shape the microbial community structures in the rhizosphere. These compounds, including sugars, amino acids and organic acids, have various effects on bacterial community composition and microbiome assembly [58]. In addition, aromatic plants are also known to be able to secrete secondary metabolites in their root exudates [59,60,61]. Due to their anti-microbial properties, some of these metabolites probably influence bacterial richness and diversity by targeting specific communities [62]. Some compounds may therefore affect microbial communities in the rhizosphere [60]. This is especially true for mono- and sesquiterpenes found in aromatic plants [60].

We noted an increase in the genus *Streptomyces* under vegetated conditions, a genus that has been described for its abilities as plant growth-promoting bacteria, but also for TE biosorption, making it particularly interesting in a phytostabilization context [63,64]. Moreover, the relative abundance of some genera (*Sphingomonas*, *Gemmatimonas*, *Kribbella*, …) was very low, but they could play an important role in bio-geochemical cycles [65,66]. In fact, rare taxa are more modulated in abundance than dominant taxa [66], and therefore could be important factors in the resilience and adaptability of soil microorganisms [66]. Interestingly, some bacterial phyla have shown the ability to limit the accumulation of TE in plants. Indeed, some *Actinobacteria* and *Proteobacteria* can alter the bioavailability of heavy metals through redox, precipitation or sorption and desorption reactions [49]. These phyla, such as *Firmicutes* and *Bacteroidetes*, are known to possess TE-tolerance genes encoding proteins involved in efflux and sequestration of TE ions, such as ATPases, D-protein or B-protein (copper tolerance and transport, respectively) and oxidative stress attenuation proteins, such as superoxide dismutase, alkyl hyperoxide reductase or mycothiol reductase [67,68].

Therefore, coriander in TE-polluted soil activated the resilience of bacterial communities. This resilience of microbial communities and functions in TE-polluted soils is usually related to a shift from sensitive to tolerant species, genetic modifications leading to TE-resistance, a transfer of genes encoding resistance or tolerance to TE, or a decrease in TE-bioavailability [69]. The recruitment of bacterial species belonging to the genera *Pseudarthrobacter*, *Arthrobacter* and *Streptomyces* by root exudates of coriander to the detriment of *Gemmatimonadota* bacteria, was associated with an increase in soil functionality that corresponds to such a shift. In this case, species belonging to *Gemmatimonadota* could be considered as indicator species for resistance to TE. Similarly, species belonging to *Pseudarthrobacter*, *Arthrobacter* and *Streptomyces* could be more sensitive to TE, but can characterize the resistance of TE-polluted soils when suitable substrates are available, such as those provided by root exudates in phytomanagement.

## 4. Materials and Methods

### 4.1. In Situ Experimental Design 

Study Area.

The study took place on an agricultural soil, historically polluted by TE (50°25′55.5″ N, 3°02′25.5″ E; elevation 23 m) in the “Hauts de France” region (northern France). The experimental site was located 600 m away from a former Pb and Zn smelter, Metaleurop Nord, whose activity generated significant amounts of TE-rich dust, resulting in TE-pollution of the topsoil (0–30 cm) [70,71]. The topsoil is characterized as silt loam, with a slightly alkaline water pH (7.9). It contains high TE total concentrations of Cd, Pb and Zn (7, 394 and 443 ppm, respectively), of which the NH_4_NO_3_ extractable fractions account for 0.114, 0.078, and 0.930 ppm, respectively [6]. These concentrations were approximately 17, 11 and 6-fold higher, respectively, than the regional background values for agricultural soils [72].

### 4.2. Experimental Design

The *in situ* experimental design was divided into 20 equal plots of 9 m^2^, to investigate the effects of one biological (AMF) and two mineral amendments on soil function, with or without planting an annual herbaceous aromatic plant species, coriander (*Coriandrum sativum* L.). The mineral amendments, ladle and oven slags, were by-products of steel production, often used in agriculture due to their buffering capacity [73]. Their main constituent was CaO and Fe_2_O_3_, respectively. The biological amendment was the commercial mycorrhizal inoculum SYMBIVIT^®^ (INOCULUMplus, Bretenière, France), which contained six AMF species: *Rhizophagus irregularis*, *Funneliformis mosseae*, *Glomus microaggregatum*, *Glomus claroideum*, *Claroideoglomus etunicatum* and *Funneliformis geosporum*. The surface of each test area was 9 m^2^ (3 m × 3 m), and each test area was randomly arranged and separated from each other by a 1 m wide corridor. A total of 20 conditions were designed (Appendix A). Before sowing, the soil of the test plots was amended or not with 2 different concentrations of 1.5 and 45 t ha^−1^ mineral amendment (ladle and oven slags) and/or the biological amendment at a rate of 10 kg ha^−1^. These plots were then sown (7.5 kg ha^−1^) or not (unvegetated condition) with coriander (seeds were provided by Iteipmai, France) in March 2019. The unvegetated plots were kept in this condition by regular manual weeding.

### 4.3. Sample Collection

At the beginning of the *in situ* experiment, 5 soil samples representing the original condition were taken with a soil auger in the 0–20 cm soil horizon, and stored at −20 °C until analysis. Three weeks after sowing, germination rate was assessed by counting the number of coriander plants in 3 random squares (1 m^2^) for each vegetated plot. After 11 weeks of cultivation, 5 new soil samples were collected in each plot from the 0–20 cm soil horizon and stored at −20 °C until analysis; the aboveground biomass and roots of 5 plants were also harvested from the vegetated area. The height of the aerial parts was measured first, before the plants were dried and weighed, while the fresh roots were washed and stored at 4 °C until stained for AMF root colonization determination. The roots were cleared in 10% (*w*/*v*) KOH and stained with 0.05% (*w*/*v*) trypan blue (Alfa Aesar) [74] to determine root colonization [75]. Briefly, root fragments were placed on microscopic slides and mycorrhizal structures (arbuscules and vesicles) were counted for each fragment observed. Three counts per fragment were performed. The percentage of total colonization was equal to the number of intersections with mycorrhizal roots × 100/number of total intersections. This percentage of colonization was expressed as the percentage of arbuscules, vesicles and total colonization.

### 4.4. Fatty Acid Analysis

After removing plant debris from the soil samples, 3 g of freeze-dried soil (3 samples per test area) were used for the analysis of fatty acid content. Lipid extraction was performed according to Frostegård et al., with a mixture of chloroform:methanol:citrate buffer—0.15M, pH 4.0—(1:2:0.8 *v*/*v*/*v*), under agitation for 2 h [76]. The lipid material was fractionated on Solid Phase Extraction (SPE) columns containing silica (6 mL volume, 500 mg sorbent, Interchim, Montluçon, France) by successive elutions with chloroform, acetone and methanol (1:2:1, *v*/*v*/*v*) into neutral lipids (NLFA), glycolipids and polar lipids containing phospholipid (PLFA) [76]. The NLFA and PLFA were trans-esterified with 0.2 M KOH in methanol to obtain free fatty acid methyl esters. The resulting fatty acid methyl esters were analyzed using a gas phase chromatography-mass spectrometer (GC-MS) Shimadzu QP-2010 Ultra (Shimadzu, Japan) equipped with a single quadrupole mass detector and simultaneously coupled with a flame ionization detector (FID). Samples were analyzed in split mode (80:1 ratio) on a ZB-1MS fast capillary column (10 m length × 0.1 mm inner diameter × 0.1 µm phase thickness, 100% dimethylpolysiloxane, Zebron, Phenomenex, Torrance Calif, CA, USA) using helium as the carrier gas at a constant linear velocity (40 cm s^−1^). The injector temperature was 280 °C, and the detector temperatures were 330 °C and 280 °C for FID and for the ion source, respectively. The temperature program started with an initial temperature of 175 °C and increased by 25 °C every minute to reach a final temperature of 275 °C, which was maintained for 0.5 min. The ionization mode was electronic impact at 70 eV and the mass range between 50.0 and 400.0 u was scanned. The single impact monitoring (SIM) mode was used simultaneously. Quantification of fatty acids was performed using nonadecanoic acid methyl ester (C19:0, Sigma-Aldrich) as an internal standard. Fatty acids were identified by comparing their relative retention time with that of commercial fatty acid methyl ester standards (C4-C24:1, Sigma-Aldrich), and by comparing them with spectra either obtained from commercial standards and/or published in the literature (NIST Standard Reference Database). Phospholipid fatty acid (PLFA) i15:0, a15:0, i16:0, i17:0, a17:0 and PLFA cy17:0, C18:1ω7 and cy19:0 were used as indicators of Gram-positive and negative bacterial biomasses, respectively [77].

### 4.5. DNA Extraction

Genomic DNA was extracted in triplicate directly from 250 mg fresh soil (*n* = 60) using the NucleoSpin Soil^®^ kit (Macherey-Nagel, Düren, Germany) according to the manufacturer’s instructions. The quality of the extracted DNA was checked using 1% (*w*/*v*) agarose gels. Quantification of the extracted DNA was performed using a SpectraMax^®^ iD3 spectrophotometer (Molecular Devices LLC, Sunnyvale, CA, United States). The extracted DNA was stored at −20 °C until use.

Polymerase Chain Reaction and sequencing of bacterial 16S rRNA gene Hypervariable regions V3–V4 of the *16S rRNA* gene were amplified from 1 ng of genomic DNA using a PCR thermal cycler (Agilent Surecycler 8800) with the forward primer CS1_341_F (ACACTGACGACATGGTTCTACACCTACGGGNGGCWGCAG) and the reverse primer CS2_805_R (TACGGTAGCAGAGACTTGGTCTCTGACTACCAGGGTATCTAATC) [78]. Two independent PCR reactions were performed per DNA sample.

### 4.6. Illumina MiSeq Sequencing

All PCR products were pooled (*n* = 60) and sent to the Genome Quebec Innovation Centre (Montreal, QC, Canada) for sequencing, using Illumina MiSeq platform, producing paired-end 2 bp × 300 bp.

### 4.7. Data Analysis–Bioinformatic Processing

Sequences were loaded into the Galaxy instance (v.2.3.0) of the Genotool bioinformatics platform (http://sigenae-workbench.toulouse.inra.fr, accessed on 27 December 2022) to be processed in the pipeline FROGS (Find Rapidly OTU with Galaxy Solution, [79]). Sequences with ambiguous bases (N) or without the specific primers were removed. The Cutadapt software [80] was used to search and cut primer sequences with less than 10% difference. The number of sequences identified was 2,660,014 and sequence clustering was performed using the SWARM algorithm (v2.1.5; [81]. A first denoising step was performed to build very fine clusters with minimal differences (d = 1), and a second one with an aggregation distance of 3. The resulting representative sequences for each cluster, or OTUs (Operational Taxonomic Units), were subjected to chimera detection using the VSEARCH algorithm [82] and eliminated, as well as those with an abundance <0.001% of the total abundance. In the end, 48,251 OTUs were retained, corresponding to 2,264,807 sequences. The taxonomic classification of each OTU against the Silva database (v138.1) was done using RDPClassifier [83]. The abundance of the different samples ranged from 65,070 to 16,243. As a result, a rarefaction of OTUs at species level was performed with 16,243 as the rarefaction value (“rrarefy” function of the vegan package). Following this, diversity indices were determined. Comparisons of means were made (Figures Shannon, and Chao1). As the Shapiro–Wilk group tests for the different indices indicated significant *p*-values (samples not following a normal distribution), comparisons via Wilcoxon–Mann–Whitney and Kruskal–Wallis tests were performed.

### 4.8. Nucleotide Sequence Accession Numbers

The 16S rRNA gene sequences of the whole dataset have been deposited in NCBI Sequence Read Archive (SRA) database and can be found under accession number PRJNA918629.

### 4.9. Soil Bacterial Metabolic Profiles

The metabolic potential of soil communities was determined from community-level physiological profiles (CLPPs) using Biolog EcoPlates (Biolog Inc.; [55]). For each soil sample, an EcoPlate containing 31 different carbon sources (and one blank with no carbon source) in triplicate was inoculated, and incubated for 196 h at 25 °C in an OmniLog^®^ system (Biolog Inc.). The rate of carbon source utilization was recorded by the reduction of tetrazolium, a redox color indicator that changes from colorless to purple and is detected at a wavelength of 590 nm. Data were recorded every 15 min of incubation and saved in OmniLog^®^ units generated by Biolog Data Analysis software (v1.7). Values for each well were calculated by subtracting the blank values from each well of the plate. A single absorbance time point at 50 h was used for the comparisons, as recommended in [84]. The metabolic potential of the soil microbial communities in each sample, expressed as the average well color development (AWCD), was calculated at the determined time point by dividing the sum of the optical density data by 31 (number of substrates). The total number of wells in a replicate with an absorbance above 25 OmniLog^®^ units was counted to determine the functional richness of the soil microbial community.

### 4.10. Statistical Analysis

Statistical analyses were performed using XLSTAT software (v2021.1.1; Addinsoft, Paris, France). Normality of the data was first subjected to the Shapiro–Wilk test. Depending on the result of this test, parametric (ANOVA) or non-parametric (Kruskal–Wallis) tests were used, followed by Fisher’s post-hoc test (Least Significant Difference, LSD). Data expressed as percentages (AMF colonization rate) were converted to arcsine values (ASIN function in Microsoft^®^ Excel v16.50) before statistical analysis. Bacterial community analysis was performed using the R environment (v4.2.2; http://www.r-project.org/, accessed on 27 December 2022). Permutational multivariate analysis of variance (PERMANOVA), based on the Bray–Curtis dissimilarity, were performed with 1000 permutations using the “adonis” function of the vegan [85] R package. Bacterial richness (Chao1) and diversity (Shannon) indexes were calculated using the Phyloseq (v1.36.0; [86]) R package.

## 5. Conclusions

This study investigated the effect of coriander cultivation and introduction of AMF in combination with mineral amendments (steel slags) on the bacterial function of the rhizosphere, an aspect that is poorly understood and still poorly studied. The introduction of amendments, such as steel slags or mycorrhizal inoculum, into the soil had no significant effect on coriander growth. However, these amendments altered the structure of bacterial communities in the rhizosphere, without affecting their metabolic potential. In fact, *Actinobacteria* were significantly less abundant under the slag-amended conditions (55% on average), while the relative proportion of *Gemmatimonadota* increased (10% on average). In contrast, soil vegetation by coriander increased Gram-negative bacterial biomass (mean 2.96 to 3.87 µg g^−1^ of DW soil) and the proportion of *Actinobacteria* phylum (62%). Thus, the vegetation improved soil quality and health by increasing the diversity of bacterial metabolic functions and diversity. Our results showed that coriander activated the resilience of bacterial communities in TE-polluted soils. Consequently, our results suggest that steel slag, which contributes to the circular economy through its recycling, in combination with soil vegetation that includes aromatic plants such as coriander, potentially provides long-term ecological benefits to TE-polluted soils and should be considered as a reliable tool for future TE-polluted soil remediation. Further studies are needed to confirm the contribution of steel slag to assisted TE-phytostabilization and its persistence in soil. 

## Figures and Tables

**Figure 1 plants-12-00618-f001:**
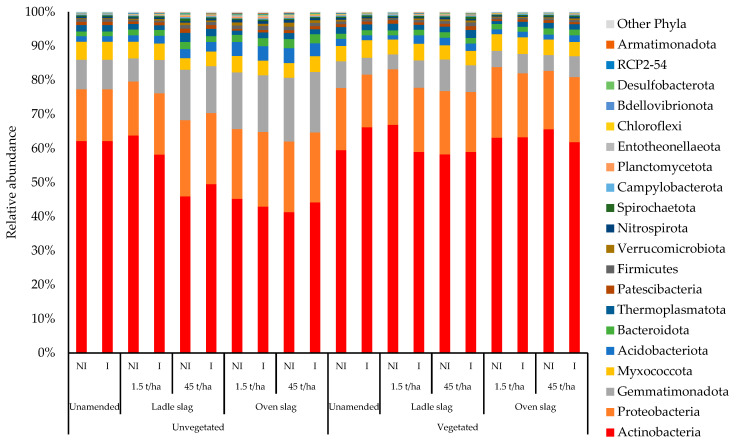
Relative abundance of bacterial operational taxonomic units (OTUs) grouped at phylum taxonomic rank in soil. NI, non-inoculated; I, inoculated.

**Figure 2 plants-12-00618-f002:**
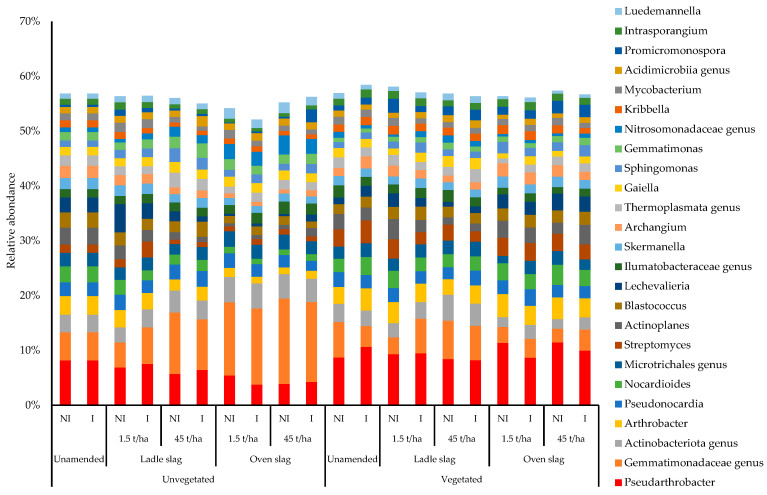
Abundances of the 25 most represented bacterial genus in soil. NI, non-inoculated; I, inoculated.

**Table 1 plants-12-00618-t001:** Growth parameters and mycorrhizal rate of coriander. Data are presented as means ± standard error. The means were obtained from five replicates (*n* = 5) for height and dry weight of plants, and three replicates (*n* = 3) for mycorrhization rate. Means followed by the same letter are not significantly different, by one-way ANOVA comparison and post-hoc Fisher test (α = 0.05).

Amendment	Dose (t ha^−1^)	Inoculation	Plant Aboveground Height (cm)	Dry Weight (g)	Mycorrhizal Rate (%)
**Unamended**		NI	69.6 ± 7.0 ab	4.1 ± 1.1 bcd	40.8 ± 10.7 abc
		I	71.6 ± 2.5 ab	4.7 ± 0.6 cd	35.6 ± 4.9 abc
**Ladle slag**	1.5	NI	68.0 ± 4.0 ab	7.0 ± 1.4 abc	31.7 ± 0.8 c
		I	65.0 ± 2.5 ab	3.2 ± 0.3 d	35.8 ± 1.8 abc
	45	NI	75.0 ± 9.8 a	8.7 ± 2.1 a	33.1 ± 3.9 bc
		I	72.7 ± 4.0 ab	5.1 ± 1.1 abcd	37.0 ± 1.3 abc
**Oven slag**	1.5	NI	70.0 ± 5.1 ab	6.1 ± 1.4 abcd	37.5 ± 4.1 abc
		I	69.0 ± 4.3 ab	4.9 ± 0.9 bcd	51.0 ± 4.6 a
	45	NI	73.8 ± 4.1 a	8.0 ± 0.7 ab	48.6 ± 7.5 ab
		I	62.0 ± 2.6 b	6.0 ± 1.9 abcd	47.9 ± 4.3 ab
**Dose**			NS	NS	NS
**Amendment**			NS	NS	**
**Mycorrhizal inoculation**			NS	*	NS
**Dose * Amendment**			NS	NS	NS
**Dose * Mycorrhizal inoculation**			NS	NS	NS
**Amendment * Mycorrhizal inoculation**			NS	NS	NS
**Dose * Mycorrhizal inoculation * Amendment**			NS	NS	NS

Significance: NS: not significant; *: *p* < 0.05; **: *p* < 0.01; according to multivariate ANOVA in bold. NI: not inoculated; I: inoculated.

**Table 2 plants-12-00618-t002:** Influence of amendments (ladle slag and oven slag used at 1.5 and 45 t ha^−1^), inoculation and vegetation on Gram-positive, Gram-negative and total bacterial biomass (µg g^−1^ of dry weight (DW) soil), and Gram-positive/Gram-negative ratio in trace element-polluted soil. Data are presented as means ± standard error. The means were obtained from three replicates (*n* = 3). Means followed by the same letter are not significantly different, by one-way ANOVA comparison and post-hoc Fisher test (α = 0.05).

State	Amendment	Dose (t ha^−1^)	Inoculation	Gram-Positive Biomass µg g^−1^ of DW Soil	Gram-Negative Biomass µg g^−1^ of DW Soil	Total Bacteria µg g^−1^ of DW Soil	Gram-Positive/Gram-Negative Bacteria
**Unvegetated**	Unamended		NI	3.64 ± 3.64 abcd	2.96 ± 1.58 abc	6.60 ± 5.20 abc	1.02 ± 0.60 bcd
			I	5.50 ± 4.26 abcd	2.85 ± 3.21 bc	8.35 ± 7.34 abc	2.93 ± 2.39 a
	Ladle slag	1.5	NI	2.43 ± 0.57 d	2.31 ± 0.94 c	4.74 ± 0.40 c	1.30 ± 0.92 bcd
			I	4.30 ± 0.95 abcd	4.11 ± 0.95 abc	8.41 ± 1.88 abc	1.05 ± 0.09 bcd
		45	NI	4.08 ± 1.76 abcd	3.58 ± 0.85 abc	7.65 ± 1.76 abc	1.22 ± 0.64 bcd
			I	2.05 ± 1.44 d	2.82 ± 0.86 bc	4.88 ± 1.85 c	0.76 ± 0.47 cd
	Oven slag	1.5	NI	6.97 ± 1.35 abc	2.86 ± 0.77 bc	9.83 ± 5.86 abc	2.42 ± 0.65 ab
			I	2.54 ± 1.13 bcd	2.51 ± 1.41 bc	5.05 ± 2.45 c	1.17 ± 0.49 bcd
		45	NI	4.70 ± 0.86 abcd	2.82 ± 0.91 bc	7.52 ± 0.08 abc	1.87 ± 0.98 abcd
			I	3.26 ± 1.81 bcd	2.82 ± 0.77 bc	6.08 ± 2.57 abc	1.09 ± 0.42 bcd
**Vegetated**	Unamended		NI	3.21 ± 2.40 bcd	2.20 ± 1.15 c	5.41 ± 3.54 c	1.35 ± 0.40 abcd
			I	2.42 ± 0.71 cd	4.06 ± 1.29 abc	6.47 ± 1.94 abc	0.60 ± 0.09 d
	Ladle slag	1.5	NI	3.65 ± 1.96 abcd	5.30 ± 0.36 a	8.96 ± 1.63 abc	0.71 ± 0.43 cd
			I	7.30 ± 3.49 ab	4.82 ± 1.98 ab	12.12 ± 5.45 a	1.47 ± 0.20 abcd
		45	NI	1.77 ± 0.35 d	2.31 ± 0.77 c	4.08 ± 1.12 c	0.78 ± 0.09 cd
			I	3.97 ± 3.09 abcd	4.37 ± 2.89 abc	8.34 ± 5.95 abc	0.88 ± 0.18 bcd
	Oven slag	1.5	NI	7.34 ± 6.21 ab	4.83 ± 1.05 ab	12.16 ± 5.49 a	1.72 ± 1.75 abcd
			I	8.14 ± 4.66 a	3.78 ± 0.34 abc	11.92 ± 4.47 ab	2.21 ± 1.29 abc
		45	NI	3.21 ± 1.92 bcd	4.15 ± 2.18 abc	7.36 ± 3.15 abc	0.91 ± 0.80 bcd
			I	2.83 ± 0.59 bcd	2.86 ± 0.78 bc	5.70 ± 0.81 bc	1.05 ± 0.35 bcd
**Vegetation**				NS	*	NS	NS
**Amendments**				NS	NS	NS	NS
**Dose**				*	NS	*	NS
**Mycorrhizal inoculation**				NS	NS	NS	NS
**Vegetation * Amendment**				NS	NS	NS	NS
**Vegetation * Dose**				NS	NS	*	NS
**Vegetation * Mycorrhizal inoculation**				NS	NS	NS	NS
**Mycorrhizal inoculation * Amendments**				NS	NS	NS	NS
**Mycorrhizal inoculation * Dose**				NS	NS	NS	NS
**Vegetation * Mycorrhizal inoculation * Amendments**				NS	NS	NS	NS

Significance: NS: not significant; *: *p* < 0.05; according to multivariate ANOVA in bold. NI: non-inoculated; I: inoculated.

**Table 3 plants-12-00618-t003:** Influence of amendments (ladle slag and oven slag used at 1.5 and 45 t ha^−1^), inoculation and vegetation on bacterial richness and diversity indexes. Data are mean ± SD (*n* = 3) for each condition. Means followed by the same lowercase letter are not significantly different, by one-way ANOVA comparison and post-hoc Fisher test (α = 0.05).

State	Amendment	Dose (t ha^−1^)	Inoculation	Chao1	Shannon
**Unvegetated**	Unamended		NI	298.7 ± 28.5 cd	4.3 ± 0.1 a
			I	305.1 ± 48.8 cd	4.4 ± 0.1 a
	Ladle slag	1.5	NI	333.3 ± 53.7 abc	4.5 ± 0.1 a
			I	318.3 ± 13.5 bcd	4.5 ± 0.1 a
		45	NI	288.1 ± 11.1 a	4.5 ± 0.1 a
			I	343.1 ± 17.7 abc	4.5 ± 0.0 a
	Oven slag	1.5	NI	335.5 ± 31.8 abc	4.4 ± 0.0 a
			I	329.9 ± 16.7 abcd	4.4 ± 0.0 a
		45	NI	312.5 ± 19.3 bcd	4.3 ± 0.2 a
			I	334.1 ± 28.6 abc	4.4 ± 0.3 a
**Vegetated**	Unamended		NI	321.7 ± 25.7 abcd	4.4 ± 0.2 a
			I	324.0 ± 23.4 abcd	4.4 ± 0.0 a
	Ladle slag	1.5	NI	320.2 ± 1.6 bcd	4.5 ± 0.1 a
			I	365.4 ± 27.9 a	4.4 ± 0.0 a
		45	NI	335.4 ± 28.2 abc	4.5 ± 0.1 a
			I	302.8 ± 13.3 cd	4.5 ± 0.0 a
	Oven slag	1.5	NI	350.7 ± 26.9 ab	4.4 ± 0.1 a
			I	310.2 ± 5.0 bcd	4.5 ± 0.1 a
		45	NI	335.7 ± 33.5 abc	4.4 ± 0.1 a
			I	340.9 ± 24.3 abc	4.5 ± 0.1 a
**Vegetation**				NS	NS
**Amendments**				NS	NS
**Dose**				NS	NS
**Mycorrhizal inoculation**				NS	NS
**Vegetation * Amendment**				NS	NS
**Vegetation * Dose**				NS	NS
**Vegetation * Mycorrhizal inoculation**				NS	NS
**Mycorrhizal inoculation * Amendments**				NS	NS
**Mycorrhizal inoculation * Dose**				NS	NS
**Vegetation * Mycorrhizal inoculation * Amendments**				NS	NS

Significance: NS: not significant; *: *p* < 0.05; according to multivariate ANOVA. NI: non-inoculated; I: inoculated.

**Table 4 plants-12-00618-t004:** Richness and diversity indexes of the 20 soil modalities. Data are mean ± SD (*n* = 3) for each condition. Means followed by the same lowercase letter are not significantly different, by one-way ANOVA comparison and post-hoc Fisher test (α = 0.05).

State	Amendment	Dose	Inoculation	AWCD	Functional Richness
**Unvegetated**	Unamended		NI	100.27 ± 18.70 ef	23.00 ± 1.63 cde
			I	109.00 ± 16.49 bcdef	23.50 ± 2.65 bcde
	Ladle slag	1.5	NI	101.51 ± 29.86 ef	21.50 ± 4.43 de
			I	131.28 ± 21.44 abcd	25.25 ± 1.71 abc
		45	NI	105.58 ± 5.46 def	24.50 ± 1.00 bcd
			I	99.88 ± 17.08 ef	21.50 ± 2.65 de
	Oven slag	1.5	NI	107.53 ± 15.34 cdef	23.75 ± 1.26 bcde
			I	98.86 ± 18.16 ef	21.00 ± 2.16 e
		45	NI	87.99 ± 23.07 f	20.75 ± 2.87 e
			I	96.76 ± 14.04 ef	21.00 ± 2.83 e
**Vegetated**	Unamended		NI	134.96 ± 6.02 ab	26.50 ± 1.91 ab
			I	148.19 ± 20.41 a	26.50 ± 1.29 ab
	Ladle slag	1.5	NI	141.52 ± 15.20 a	25.75 ± 0.50 ab
			I	136.81 ± 25.96 a	26.75 ± 1.50 ab
		45	NI	128.84 ± 25.97 abcd	25.00 ± 3.65 abc
			I	122.11 ± 19.30 abcde	25.50 ± 2.52 abc
	Oven slag	1.5	NI	135.67 ± 18.88 ab	25.75 ± 1.71 abc
			I	133.32 ± 19.25 abc	25.25 ± 3.30 abc
		45	NI	147.66 ± 12.12 a	28.00 ± 1.63 a
			I	145.17 ± 24.64 a	26.75 ± 1.89ab
**Vegetation**				***	***
**Amendments**				NS	NS
**Dose**				NS	NS
**Mycorrhizal inoculation**				NS	NS
**Vegetation * Amendments**				*	NS
**Vegetation * Dose**				NS	NS
**Vegetation * Mycorrhizal inoculation**				NS	NS
**Mycorrhizal inoculation * Amendments**				NS	NS
**Mycorrhizal inoculation * Dose**				NS	NS
**Vegetation * Mycorrhizal inoculation * Amendments * Dose**				NS	NS

Significance: NS: not significant.*: *p* < 0.05; ***: *p* < 0.001 according to multivariate ANOVA. NI: non-inoculated; I: inoculated.

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
