# Peer review of "Coriander (Coriandrum sativum) Cultivation Combined with Arbuscular Mycorrhizal Fungi Inoculation and Steel Slag Application Influences Trace Elements-Polluted Soil Bacterial Functioning"

_plants, 2023, doi:10.3390/plants12030618_

Round 1

Reviewer 1 Report

Ø  Abstract: Adequate and nicely written

Ø  Introduction is adequate and written nicely.

Ø  Materials and Methods:

Methodology section has been described well; however, some improvements may increase the soundness:

·  Experimental design and execution section need to clearer about the treatment and their execution.

·      Line 421 is incomplete/missing.

·      Line 456: include the information about the nucleotides database.

·     Methodological part of Illumina MiSeq and Nucleotide Sequencing need to be clearer.

·      Statistical Analysis section should be last part of methodology.

Ø  Results: Results are of present investigation are written clearly. However, it is needed to be clear about the significance of alphabets next to the data presented in the table.

Table-1: As in the writeup, effect of management is non-significant on the plant height but alphabets (a) and (b) should be significantly varying. Please make it clear.

As in table-1 & table-2 the information about the replicates is given alongside table title while in next tables (table-3 and 4) its given at the end. May follow the uniform pattern.

Table number 3 has been replicated twice, instead of writing the table-4. Please revisit.

Ø  Discussion: results of the present investigation have been written adequately and discussion is supported with scientific factual reasoning.

Ø  Conclusion is written effectively however, concluded study can be improved factually/quantitatively considering the dataset/results of present investigation.

Ø  References: all the references included in the manuscript are relevant. Although, in the reference like 83, Journal name is missing.

Author Response

Reviewer 1 :

Materials and Methods:

Methodology section has been described well; however, some improvements may increase the soundness:

Experimental design and execution section need to clearer about the treatment and their execution.

Response: We added a table (Table S1, line 511) to clarify the experimental design and the treatments used.

Line 421 is incomplete/missing.

Response: The modification was done, line 415 ”Polymerase Chain Reaction and sequencing of bacterial 16S rRNA gene”.

Line 456: include the information about the nucleotides database.

Response: The modification was done line 451: accession number PRJNA918629.

Methodological part of Illumina MiSeq and Nucleotide Sequencing need to be clearer.

Response: We have modified the two paragraphs “polymerase chain reaction“ lines: 416-420, and “Illumina MiSeq Sequencing” lines: 423-425.

Statistical Analysis section should be last part of methodology.

Response: The modification was done, the section “Statistical analysis” is now beginning line 473.

Results: Results are of present investigation are written clearly. However, it is needed to be clear about the significance of alphabets next to the data presented in the table.

Response: Different letters indicate significant differences according to the one way ANOVA and post-hoc Fisher test. The modifications have been included in the tables on lines 114-115, 133-134, 160-161, 200-201, and 515-516.

Table-1: As in the writeup, effect of management is non-significant on the plant height but alphabets (a) and (b) should be significantly varying. Please make it clear.

Response: These are two different tests, different letters indicate significative differences according to one way ANOVA, and “*” according to a multivariate ANOVA, which is more sensitive because it pools conditions and therefore has a larger sample size (n). We have indicated this more clearly in the tables captions.

As in table-1 & table-2 the information about the replicates is given alongside table title while in next tables (table-3 and 4) its given at the end. May follow the uniform pattern.

Response: Thanks for your comment, the modification was done.

Table number 3 has been replicated twice, instead of writing the table-4. Please revisit.

Response: We have corrected this mistake in line 199.

Conclusion is written effectively however, concluded study can be improved factually/quantitatively considering the dataset/results of present investigation.

Response: The modification was done lines 497-504.

References: all the references included in the manuscript are relevant. Although, in the reference like 83, Journal name is missing.

Response: the missing reference has been added line 744.

Reviewer 2 Report

Coriander (Coriandrum sativum) cultivation combined with arbuscular mycorrhizal fungi inoculation and steel slag application influences trace elements - polluted soil bacterial functioning.

Despite the positive aspects of this study, there are the following remarks:

L. 67-69 - the conclusion does not follow from the previous text.

L. 112 – you should write about native AMF.

You cannot use ANOVA with Mycorrhizal inoculation because the Mycorrhizal rate (%) is sometimes higher in non-inoculated sites than inoculated sites.

L. 119 – please put an indent.

L. 125 – please decipher «DW»

L. 160 - Please move Table 3 to the section Bacterial α-Diversity of Soil Biotopes.

L. 204 – you should rename Table 3 to Table 4.

Your article will be better if you indicate for which substrates there was a difference in substrate utilization.

L. 253 - «in our agricultural polluted soil» do you really mean «in our»? Did Fontaine et al study the same samples as you?

L. 358 - Table 1 indicate only 10 conditions.

L. 386- please specify in more detail according to exactly.

L. 463 - where else is the post-hoc Fisher test mentioned?

Please move section Soil bacterial metabolic profiles before section Statistical analysis.

L. 502 - Figure S1: title; Video S1: title missing from the text.

L. 597 - incomplete link.

Your article would be better if you determined the physical and chemical properties of the soil.

Author Response

Reviewer 2 :

Despite the positive aspects of this study, there are the following remarks:

  1. 67-69 - the conclusion does not follow from the previous text.

Response: We move this part on lines 77-79.

  1. 112 – you should write about native AMF.

Response: The presence of native AMF in soil has been mentioned in the text, line 113.

You cannot use ANOVA with Mycorrhizal inoculation because the Mycorrhizal rate (%) is sometimes higher in non-inoculated sites than inoculated sites.

Response: As we are working in situ in the field, it is possible that there are higher values in the non-inoculated modality than in the inoculated plots. AMF colonization rate (%) were converted to arcsine values before statistical analysis (ASIN function on Microsoft® Excel v16.50). Normality of the data were first submit to Shapiro-Wilk test, and finally parametric test (ANOVA) was used, followed by a post-hoc Fisher (Least Significant Difference-LSD) test.

  1. 119 – please put an indent.

Response: We have added an indent in line 121.

  1. 125 – please decipher «DW»

Response: the abbreviation DW has been explained in the text line 126 and in Table 2 line 134.

  1. 160 - Please move Table 3 to the section Bacterial α-Diversity of Soil Biotopes.

Response: The table is now on line 161.

  1. 204 – you should rename Table 3 to Table 4.

Response: this has been corrected in line 201.

Your article will be better if you indicate for which substrates there was a difference in substrate utilization.

Response: We have added a supplemental table (Table S3), containing the substrates data.

  1. 253 - «in our agricultural polluted soil» do you really mean «in our»? Did Fontaine et al study the same samples as you?

Response: Fontaine et al did not use the same soil samples but it was an experiment in pot with polluted soil from the same plot, contrary to our experiment which was carried out in situ. We changed the sentence line 249.

  1. 358 - Table 1 indicate only 10 conditions.

Response: we have added Table S1 mention in line 353 because we didn’t cite the good table.

  1. 386- please specify in more detail according to exactly.

Response: this sentence has been added in the text lines 380-382 : “Lipid extraction was performed according to Frostegård et al, with a mixture of chloroform : methanol : citrate buffer - 0.15M, pH 4.0 - (1 : 2 : 0.8 v/v/v), under agitation for 2 hours”.

  1. 463 - where else is the post-hoc Fisher test mentioned?

Response: This test is used to cluster the results of the one-way ANOVA of each table (Table 1-4, S2) and thus indicate by a letter the result of this clustering (different letters indicating a significant difference).

Please move section Soil bacterial metabolic profiles before section Statistical analysis.

Response: The section Soil bacterial metabolic profiles has been moved before section Statistical analysis.

  1. 502 - Figure S1: title; Video S1: title missing from the text.

Response: The modification was done, this information has been added in the text :

Table S1: Experimental design of plots describing all 20 modalities studied; Table S2: Influence of amendments and vegetation on most abundant phyla relative abundance, Table S3. Influence of amendments and vegetation (V: vegetated, UV: unvegetated, NI: not-inoculated, I: inoculated, LS: ladle slag and OS: oven slag, NA: not-amended, 1.5 and 45 t.ha-1) on Biolog substrates family.

  1. 597 - incomplete link.

Response: The modification was done:” Smith, S.; Read, D. Mycorrhizal Symbiosis; Elsevier, 2008; Vol. 3; ISBN 9780123705266.”, in line 588.

Your article would be better if you determined the physical and chemical properties of the soil.

Response: The article by Raveau et al (6) that we cited, in which the study is conducted on the same polluted plot, already deals with this aspect. This is why we did not carry out new soil analyses.

Round 2

Reviewer 2 Report

Remarks were taken into account by the authors and the manuscript is improved and now I recommend publication of this work in journal of Plants.